Location decision of low-altitude service station for transfer flight based on modified immune algorithm

Chen Huaqun 1 chqtx@126.com
Yang Weichao 1
Tang Xie 2
Yang Minghui 1
Huang Fangwei 1
Zhu Xingao 1
1 Air Traffic Management Department, Civil Aviation Flight University of China , Guanghan, Sichuan , China
2 Sichuan Highway Planning Survey Design Institute Co. LTD., Ternal , Chengdu, Sichuan , China
Shi Kaize
Electronic publication date: 2023 Nov 1
Publication date: 2023
Volume: 9
Electronic Location ID: e1624
Received 2023 Jun 30; Accepted 2023 Sep 8
Copyright: © 2023 Chen et al.
Copyright year: 2023
Copyright holder: Chen et al.
License: This is an open access article distributed under the terms of the Creative Commons Attribution License, which permits unrestricted use, distribution, reproduction and adaptation in any medium and for any purpose provided that it is properly attributed. For attribution, the original author(s), title, publication source (PeerJ Computer Science) and either DOI or URL of the article must be cited.
License URL: https://creativecommons.org/licenses/by/4.0/

Keywords: LAFSS, Multi-objective optimization, Location decision, MIA, Matlab simulation, JAVA spring, Cesiumjs map

Funding: Sichuan Science and Technology Program 2022YFG0197 Civil Aviation Flight University of China J2022-061 This work was supported by the Sichuan Science and Technology Program (NO: 2022YFG0197) and the general program of Civil Aviation Flight University of China (J2022-061). The funders had no role in study design, data collection and analysis, decision to publish, or preparation of the manuscript.

==============================
The location of Low-Altitude Flight Service Station (LAFSS) is a comprehensive decision work, and it is also a multi-objective optimization problem (MOOP) with constraints. As a swarm intelligence search algorithm for solving constrained MOOP, the Immune Algorithm (IA) retains the excellent characteristics of genetic algorithm. Using some characteristic information or knowledge of the problem selectively and purposefully, the degradation phenomenon in the optimization process can be suppressed and the global optimum can be achieved. However, due to the large range involved in the low-altitude transition flight, the geographical characteristics, economic level and service requirements among the candidate stations in the corridor are quite different, and the operational safety and service efficiency are interrelated and conflict with each other. And all objectives cannot be optimal. Therefore, this article proposes a Modified Immune Algorithm (MIA) with two-layer response to solve the constrained multi-objective location mathematical model of LAFSS. The first layer uses the demand track as the cell membrane positioning pattern recognition service response distance to trigger the innate immunity to achieve the basic requirements of security service coverage. In the second layer, the expansion and upgrading of adjacent candidate sites are compared to the pathogen’s effector, and the adaptive immunity is directly or indirectly triggered again through the cloning, mutation and reproduction between candidate sites to realize the multi-objective equilibrium of the scheme. Taking 486,000 km2 of Sichuan Province as an example, MIA for LAFSS is simulated by the MATLAB platform. Based on the Spring open source application framework of Java platform, the cesiumjs map data is called through easyui, and the visualization of site selection scheme is presented with the terrain data of Map World as the background. The experimental results show that, compared with dynamic programming and ordinary immunization, the immune trigger mode of double response and the improved algorithm of operation parameter combination designed by the Taguchi experiment, the total economic cost of location selection is reduced by 26.4%, the service response time is reduced by 25%, the repeat coverage rate is reduced by 29.5% and the effective service area is increased by 17.5%. The security risk, service efficiency and location cost are balanced. The present work is to provide an effective location method for the layout number and location of local transfer flight service stations. For complex scenes with larger scale of low-altitude flight supply and demand and larger terrain changes in the region, the above research methods can be used to effectively split and reduce the dimension.

Introduction

The location decision of Low-Altitude Flight Service Station (LAFSS) should not only consider the fixed construction and the dynamic cost of late operation, but also the uncertain factors caused by service signal coverage and transfer route. Therefore, the location optimization objective of LAFSS is a Constrained multi-objective optimization problem (MOOP) (Qian, 2017), which focus on the minimum service coverage requirement of flight safety as the basic criterion and takes into account service cost and service efficiency. IA has become one of the most commonly used methods to solve constrained optimization problems because of its unique advantages such as immune memory and clone selection (Li, Lin & Zhong, 2022).

Since 2010, China has promoted low-altitude airspace reform, and general aviation has been listed as a development focus. In the past 10 years, the average annual growth rate of aviation enterprises and aircraft has exceeded double figures. By the end of August 2022, there are 665 traditional aviation enterprises in China’s mainland, with a total of 2,807 aircraft in actual operation, 389 general airports in the country, and 14,355 UAV operating enterprises (Ma & Wang, 2023). With the development of China’s general industry, the demand for LAFSS is increasing dramatically. However, there are only 13 LAFSS under construction or already built in China, and most of them rely on general aviation enterprises or are upgraded to B-class LAFSS. It is only a small range of navigation services, lack of complete monitoring, tracking and other flight services. LAFSS is to provide full service and support for general aviation activities. The rationality of its location directly affects the safety, order and efficiency of flight mission execution.

At present, most researches focus on the strategic positioning, management mode and service function of LAFSS. For example, the construction idea of low altitude flight service (LAFS) support system is analyzed from the perspective of flight declaration (Zhang, 2020) and operation monitoring (Fang, Zhu & Wu, 2018). This article analyzes the main functions of flight service center from the perspective of navigation requirements (Liao et al., 2020). Based on information construction, the requirements of flight service software are discussed. The service technologies of low-altitude flight in European and American countries has reached an advanced level in the world. The Federal Aviation Administration of the United States has set up more than 200 manual or automatic flight service stations. Benchmarks & performance are used to review flight service station operators market (Federal Aviation Administration, 2016). Reduction of remote communications outlets used by flight service stations in the conterminous united states has been discussed (Federal Aviation Administration, 2017). However, it has not formed the theoretical system and implementation method for site selection optimization (Wang, Ma & Wang, 2015).

Layout of LAFSS for general navigation belongs to the category of facility location selection. In recent years, global scholars have conducted extended studies based on multi-objective planning theory and artificial intelligence technology (Zhang et al., 2022). For example, multi-objective programming method (Masashi, 2021) is used to optimize facility location, but the algorithm and model mainly target at discrete local services. Dynamic changes in demand of site selection optimization is considered in hospitals location-allocation (Shang et al., 2023), facilities location (Li, 2021) and simultaneous pickup (Xu, Zheng & Deng, 2020), which made the site selection robust. The above researches are mainly aimed at plane location, and do not take into account the signal errors caused by ground obstruction, stability of navigation facilities and performance differences of airborne equipment during flight, which cannot reflect the spatial effectiveness of actual flight services.

A modified genetic algorithm (Chen, Xiong & Huang, 2021) was used to solve the location of multi-target LAFSS with discrete aggregation requirements. However, the genetic algorithm is relatively fixed in crossover and mutation operators, resulting in less flexibility in solving optimization problems. Aiming at these defects, the concept and theory of immunity (Wang, 2017) were applied to the genetic algorithm (Liu, 2021), and a popular multi-target Immune Algorithm (IA) is formed. The basic model of the immune network states that the immune system (Sun, 2021) consists of a network of cellular molecular adjustments that recognize each other even in the absence of antigens. The principle of immune network is applied in the field of multi-objective optimization. The discrete immune network model (Zhong et al., 2021) points out that antibodies are recognized according to the Euclidean distance, and the closer the distance is, the greater the similarity between two antibodies (Cheng & Cui, 2020). Subsequently, in order to improve the performance of multi-objective IA in solving MOOP (Cao et al., 2018), a large number of multi-objective IA have been proposed, such as using IA to optimize site location planning and capacity expansion (Jiang & Hao, 2020). Therefore, this article will use IA not only to retain the good genes of the original parents of genetic algorithm (Yao, 2021), but also to use immune factors selectively and purposefully to obtain more characteristic information or knowledge of other sites with candidate conditions by taking advantage of the characteristics of antibody cloning selection (Sun, 2021) and automatic antigen recognition in the immune system, so as to inhibit degradation in the process of site selection optimization.

Low altitude flight in China is a new industry for specific object services. Due to the constraints of airspace, budget or terrain, flight services are usually carried out with limited facilities to ensure that as many demand areas in the space are covered. Based on the “Overall Plan for the Construction of Low-altitude Flight Service Guarantee System” (CAAC, 2018), this article obeys on flight safety as the primary criterion, while taking into account service efficiency and user convenience as the orientation. These airspace where transfer flights are permitted are defined as the demand for low altitude traffic services with motion equation. Setting coverage theory is applied to built the limitation with resource constraint and service response. A constrained multi-objective optimization location mathematical model is established. The Modified Immune Algorithm (MIA) with two-layer response is used to solve the model. The optimal balanced site layout scheme is determined. The feasibility and superiority of the model and algorithm are verified by an actual example. MIA is proposed firstly to solve the problem of flight service location decision for low altitude navigation in China. The visualization of location decision is realized by using artificial intelligence algorithm and computer graphics technology.

Problem analysis and hypothesis

Problem description

According to Guiding Opinions on Construction and Management of the General Aviation Flight Service Station System (CAAC, 2012), the LAFSS is divided into four classes which are nation, region and Class A and B, namely L = {1,2,3,4}. Since the national and regional administrative establishment has been completed, the grade L of LAFSS selected in this article is three or four. Each provincial administrative region meets the requirements of setting up one to three flight service stations of Class A in principle and several Class B as needed. Referring low altitude collaborative management, the categories of service demand in low-altitude flight is divided into two types, namely D = {1,2}, which represents demand in field area or transition corridor respectively. Each demand object has three service coverage modes, namely C = {0,1,2}, which represents independence, joint and repeat coverage respectively. The required service overlay is shown in Fig. 1.

Figure 1 Demand coverage diagram.

Demand I is a collection of demands from U1 city and H2 county, but taken into account the provincial level of P1, which adopts the regional 1st level and separate coverage mode. Demand II relies on urban C2, which adopts the Class A mode and separate coverage. Demand III from county H3 and the navigable town t3 belongs to the transfer corridor, which adopts the Class A and joint service coverage mode. Demand IV is part of local navigable town t2, which is divided into Class A and B and joint coverage mode.

If the service distance or response time from any flight demand area to the candidate site is less than the safety requirements Ø, the demand area is covered by the candidate site. Since the flight service mode may be independence, joint or repeat coverage, the service coverage level function of low-altitude flight demand based on the coverage theory (Yao, 2021) is as follows in Eq. (1).

(1) CL=∑l∈Lρl∑c∈Cρc+∂

where ρl is the weight of the adopted 1st level flight service station. ρc is the weight of the service coverage method. ∂ is the service random fluctuation coefficient.

Model assumptions

Physical space requirements of low altitude flight include in-field flight and transfer corridor flight. The former is a discrete combination optimization problem, while the latter needs to carry out task relay at each service station within the range of the transfer to achieve service demand coverage within the range of flight path, which belongs to continuous decision planning. Therefore, setting coverage and median quantification (Sharifi et al., 2023), this article establishes site selection mathematical model of multi-objective equilibrium optimization with minimum economic cost, fastest service response and minimum effective service demand distribution distance.

In order to facilitate the construction of mathematical model, the following assumptions are made. If service radius is reachable, a flight service station can cover multiple flight demand areas. And

A flight demand area is served by only one station.

Different levels of flight service stations correspond to different service radius, flight heights, coverage grade and decision weights.

The same area of flight demand corresponds to different operational types, but the largest range is chosen.

The candidate stations for low-altitude flight service are generated at the demand gathering area or nearby towns, without considering random installation.

Parameter setting

Based on the flight course in low-altitude airspace, the relative parameters include in low-altitude transit demand, candidate flight service sites, siting decisions, service response and signal coverage.

There are K transit corridor flights. All the main parameters of every transit corridor are described in set. Ωd=[ω1d,ω2d...ωkd...ωKd] is a set of transfer requirements, k ∈ K.

ωkd=[Psk(x,y,z)...Ptk(x,y,z)...Pek(x,y,z),htk] is the position coordinates of any service requirement object, whose position contains start, any moment t and the end.

S = {S1, S2…Sj…, SN} is a set of candidate sites, j ∈ N. Because of low-altitude flight activity is a socially derived demand, the site selection usually relies on a navigable airport, takeoff-landing point, or a nearby town. Allowing layout located will only be at a given finite number of location points.

Sj = {oj, rj, qj, hj, Cj} is a attributes set of any each candidate, which contains center coordinates oj. service radius rj, service supply qj, service coverage height hj, and service station cost Cj. Service station cost Cj consists of land cost Cj1, facility cost Cj2, operation and maintenance cost Cj3, and scalable set-aside cost Cj4.

Three decision variables are xj, yjk and gjj+1. xj is assigned to be value 1 if service candidate station selected or 0 otherwise. When the yjk is assigned to be value 1 if demand point is covered individually, or 0 otherwise. gjj+1 is assigned to be value 1 if service overlap intersection, or 0 otherwise.

Service response. tjk is the time which low service candidate station Sj responds to demand point ωkd. tjkdepends on the service distance ljk between the geometric center of supply and demand target okd and Oj, the signal propagation speed of the communication navigation facilities adopted by each candidate station Vc, the human-machine interaction response time Ta and the signal coverage variance error Te.

Considering the influence of terrain and geomorphic changes and blocking of communication and navigation signals by ground objects, the value of tji follows σ Poisson distribution. tjk is taken according to Eq. (2).

(2) tjk=[2(ljk÷Vc)]2+Ta2+σTe2

The radio transmission speed is 3 × 108 m/s (Sakai et al., 2021), while service distance is usually between 200 and 300 km, and the signal transmission VC does not change much. However, due to the obstruction of the terrain, Ta and Te are quite different. In this article, the signal coverage errors of LAFSS in plains, hills and plateaus are respectively 1 to 3 s, i.e., Te∈[1,3], and human-machine interaction response time Ta∈[1,6].

Transition track

Service coverage analysis

Tk is the total flight time of any transfer flight path ωkd. The candidate service stations at the start are determined. With the transition flight, there may be multiple candidate service stations oj and oj′ and other service radii covering the track range. The service relay process is shown in Fig. 2.

Figure 2 Service relay diagram for transit flights.

From the start position Psk(x,y,z) to the final destination Pek(x,y,z), each service station through the whole process can be likened to be completion of the task. Therefore, the location of LAFSS under such flight requirements is optimized. In this article, the flight time is taken as the continuous interval and the service range of the discrete candidate stations around the track is taken as the coverage space to determine the best location scheme and achieve continuous decision planning.

Coordinates conversion

Cartesian coordinates are used to describe the movement process of transition flight more visually and graphically, which are easy to realize the combination of number and shape. The longitude and latitude coordinates obtained by the GIS are set as (B,L,H) (Zhou et al., 2023), and the geographic coordinates are projected to the plane coordinate system. The coordinate conversion mode is shown in Fig. 3.

Figure 3 Geographical and Cartesian coordinates conversion diagram.

The transformation equation for the Gauss–Kruger projection is shown in Eq. (3).

(3) X=(N+H)∗cos⁡LY=(N+H)∗cos⁡B∗sin⁡LZ=[N∗(1−e2)+H]∗sin⁡B}

where rmax and rmin is the longest and shortest radius. f is the ellipsoidal flatness. e is the first eccentricity which value is taken by rmax2−rmin2∖rmax. W is the first auxiliary coefficient i.e., calculated by (1−e2)B. N is the radius of curvature of the ellipsoidal surface given by rmax∖W.

Determination of transit trajectory

If calculate roughly, simple aerial linear trajectory is given out. o-xyz coordinate system should be built as geodetic plane. The plane coordinates of the origin place is (0,0,0), Vt is the velocity at any time tk in any flight path of the aircraft, and α is the angle between direction of travel and ground plane (Højlund et al., 2022). The coordinate system was created as shown in Fig. 4.

Figure 4 Simple aviation linear trajectory coordinate change.

Based on the equation of motion with direct track, the coordinates of any position Pki accelerated at any time tk are expressed as in Eq. (4).

(4) xt=(v0tk+12atk2+13btk3)cos⁡αyt=±Δytkzt=(v0tk+12atk2+13btk3)sin⁡α

However, actual low-altitude flight path consists of a combination of straight lines and turning curves. Flight speed is a nonlinear continuous change process from small to large and then decreasing to 0 (Li & Peng, 2007). Therefore, in order to improve coverage accuracy of the service requirements, three-dimensional coordinates of improving hybrid motion trajectory are established in this article, as shown in Fig. 5.

Figure 5 Coordinate projection of precise flight paths.

β is the angle between the tangential velocity and the ground plane, Vtk is the instantaneous velocity of the aircraft at time tk in the flight trajectory, Vt′ is the projection of the instantaneous velocity in the ground plane. γ is the angle between Vt′ and the positive direction of the y-axis. In the turning section, Ψ is defined to describe the angle between the plane and the horizontal plane where its circular route is located. Φ is the angle between the intersection line of the two planes and the y-axis.

Let the coordinates of initial position be (x0,y0,z0). When projected trajectory on the o-x-y plane is a straight line, coordinates of any position Pki (xt,yt,zt) at moment tk is described in Eq. (5).

(5) {xt=x0+(v0tk+12atk2+13btk3)cosβsinγyt=y0+(v0tk+12atk2+13btk3)cosβcosγzt=z0+(v0tk+12atk2+13btk3)sinβ

When the projection of trajectory on the o-x-y plane is a curve, the radius of the turn is r, the center of the circle O1 coordinates (x01,y01,z01), O1′ is the O1 projection, the radius O1C is parallel to AB, C′ is the projection of C.

Thus, the coordinates in the coordinate system at the moment tk are described in Eq. (6).

(6) {xtk=x01+XcosΦ=x01+rsin(θ1+Vtkr)cosΨcosΦytk=y01+YsinΦ=y01+rcos(θ1+Vtkr)sinΦztk=z01+rsinθ2sinΨ=z01+rsin(θ1+Vtkr)sinΨ

The precise coverage range of service requirement for kth low-altitude transition corridor in target object scope can be determined at any time tk.

Coverage model

Objective function

We assume that each candidate service site is independent, each service station can provide serve for multiple demand areas, but one demand area is only served by one candidate site. It belongs to the hierarchical setting location problem of different service levels.

According to transfer flight requirements Ωd, the most appropriate location is select to set up station Sj. At last the optimal site plan S* will be obtained which should ensure flight safety, pay out the minimum cost, obtain the maximum effective coverage area, decrease the minimum service response time and repeating coverage area. In a word, we achieve the maximum economic benefit. However, the optimization object Sj ∈ S is a discrete state in the solution space and belongs to discrete multi-objective combinatorial optimization. Main optimization objectives are shown in Eqs. (7)–(10).

If the candidate site Sj is selected, the sum of construction, operation, maintenance and subsequent scalable uncertain costs will be calculated in Eq. (7).

(7) f1(x)=∑j∈Nxj(Cj1+Cj2+Cj3+Cj4)

Based on the response time priority principle, the distance between candidate sites and flight demand is determined. And then, taking the distance over speed of the propagation of the navigation facilities, the service response time is obtained. The objective function of response time is expressed in Eq. (8).

(8) f2(x)=∑j∈Nxj∑k∈Kyjk[2(ljk÷Vc)]2+Ta2+σTe2

where yjk means that there are two or more joint coverage, then we will choose the minimum distance by comparing function in Eq. (23).

Compared the distance between the geometric center of two adjacent low altitude flight service candidate sites with service radius, the overlap area of service coverage is judged. The repeating coverage area is calculated by using transformed right-angle plane coordinates. Taking account of all the distances between adjacent candidate sites within study area, the repetitive coverage objective function is expressed in Eq. (9).

(9) f3(x)=∑j∈Nxjgjj+1(θjπ(rjl)2+θj+1π(rj+1l)2360−dj,j+1Δhj,j+1)

where dj,j+1 is the distance between the geometric centers of adjacent sites, the half-height of overlapping area is Δhj,j+1, and the corresponding circle center angles of the intersecting areas are θj and θj+1.

The effective service coverage area not only depends on the number and rank of candidate sites, but also is influenced by the repeating coverage area. The corresponding objective function is expressed in Eq. (10).

(10) f4(x)=∑j∈Nxj∑k∈K∑j∈N∑l∈Lyjkπ(rjl)2−f3(x)

The service efficiency is not only dependent on the quality and the location of selected service sites, but also is related to the service distribution scheme between sites and demand points closely.

Multi-objective model

The mathematical model of LAFSS location with the shortest total service response time, the lowest cost, the largest service coverage area and the lowest repeated coverage is established based on the multi-objective programming theory as follows in Eqs. (11)–(14).

(11) minZ1=∑l∈LLρl∑j=1N(Cj1+Cj2+Cj3+Cj4)xj

(12) minZ2=∑CL∑j=1N∑k∈Kxi∑j∈Kyjk[2(ljk÷Vc)]2+Ta2+σTe2

(13) minZ3=∑j=1N∑l∈Lxjgjj+1(θjπ(rjl)2+θj+1π(rj+1l)2360−dj,j+1Δhj,j+1)

(14) maxZ4=∑c∈Cρc∑j=1Nxj∑k∈K∑j∈Nyjkπ(rjl)2−Z3

Constraint s.t: Service response time is determined by the service coverage altitude of the flight service station and the demand altitude in Eq. (15).

(15) hkd/Vc≤tjk≤[hj/(Vc+Ta+Te)]

The service mode of the demand area is limited by the completed LAFSS in Eq. (16).

(16) ∀l∈L,c∈C,d∈D,k∈K,j∈N

The number assigned service tasks does not exceed the number of selected service stations in Eq. (17).

(17) Yjk≤Xj

Each demand point is served by one and only one service station in Eq. (18).

(18) ∑j∈N∑k∈Kyjk=1

Decision variables for site selection, service and repeat coverage are list separately from Eqs. (19)–(21).

(19) xj={0,1}

(20) yjk={0,1}

(21) gjj+1={0,1}

Goal conflict analysis

Mathematical model for site selection of LAFSS is a MOOP. The level, number and location of stations determine the four goals. There are some conflicts between objectives, such as the response time f2(x) is inversely proportional to the station cost f1(x), the level and number of stations affecting the station cost f1(x) are proportional to the service coverage area f3(x) and the repeating coverage area f4(x). However, the optimization objectives except f4(x) are as small as possible. The conflict analysis among the four objective functions is shown in Fig. 6.

Figure 6 Conflict between optimization goals.

All objectives are interrelated and in conflict with each other, it is impossible to obtain a global optimal solution that makes multiple objectives reach the optimal. Usually, we can only make balance between multiple targets in a given area.

Ia for multi-layer response

Algorithm design

Combined four objectives for service safety with efficiency constraints, this article proposes an improved two-layer response of immune optimization algorithm based on the operating mechanism of biological immune system to solve the MOOP of location decision of LAFSS. The IA has strong adaptability and robustness in solving MOOP (Jerne, 1974) and is a new intelligent optimization algorithm constructed artificially (Hajela & Lee, 1996). Compared with genetic algorithm, IA can produce a variety of specific antibodies, which has been proved to be an effective way to generate Pareto-Edge worth end faces (de Castro & Timmis, 2002).

The decision process is a kind of multi-process parallel optimization, which can obtain multiple sub-optimal solutions while seeking the optimal solution. In order to avoid the loss of the optimal equilibrium solution due to premature maturity, local adjustment of station rank and clonal reproduction of adjacent stations are used. Antibody encoding

Besides the candidate station sites are selected or not in the process of LAFSS location planning, there are also corresponding service station grades. Otherwise, this article chooses natural number coding. The IA treats the problem to be solved as the antigen which is the demand area of low-altitude flight covered by the service. The solution of the problem is the antibody, which is the best location. Each antibody corresponds to a location scheme. The value of the loci in the antibody corresponds the candidate site selected or not and the grade.

Natural number coding is applied, and the encoding length of the solution is N i.e., the number of candidate sites. The antibody code is denoted by Am=(S1l,S2l...Sjl...SNl), where j∈N,l∈L,m∈M. Am represents the situation where the location scheme corresponding to the mth antibody is selected for the jth candidate site and the service station level l corresponding to the selected site. All the situations selected are as following in Eq. (22).

(22) Sjl={4,Selectedandrankedas43,Selectedandrankedas32,Selectedandrankedas21,Selectedandrankedas40,Notselected

Generation of initial population

The generation of initial population plays an important role in optimization iteration of IA. In order to construct a feasible initial solution quickly, this article introduces the concept of clustering effect in economics, i.e., the centripetal force generated by the spatial concentration of various industries to attract economic activities to a certain region. According to the requirements of existing stations, initial candidate points of LAFSS are locked in the existing navigable airports, take-off and landing points or cities. The range of candidate points is further narrowed, the number of iterations is reduced, and the solution accuracy is increased.

Calculated track position Ptk(x,y,z) at any time t of the transition flight corridor k, the candidate station is classified as one of the initial population members when the distance dtjk is less than service radius rj. The judgment function is shown in Eq. (23).

(23) dtjk=|Oj(x,y,z)Ptk(x,y,z)|≤rj

Colony diversity

The weight method is commonly used to transform the MOOP into a single objective optimization problem (Wang, 2017), affinity evaluation function between antibodies and antigens is shown in Eq. (24).

(24) f(Sjl)=τ11f1(x)+τ21f2(x)+τ31f3(x)+τ4f4(x)

The value of τ1,τ2,τ3andτ4 is the weight coefficient of each objective function, and the sum is 1. This function can evaluate the merits of the candidate site solution. The value of the evaluation function f(Sjl)∈(0,f4(x)], the larger the value, the better the candidate site solution.

A deformable R−bit continuous method is used to calculate affinity between antibodies in Eq. (25).

(25) Sv,s(i)=kAiAm∖LAm

where LAm is the total length of the antibody code. kAiAm is the same number of coding bits for both antibodies.

Antibody concentration represents the ratio of similar antibodies among antibody groups that also means the proportion of similar solutions in the site selection of LAFSS. It can characterize the diversity of antibody population (Cheng & Cui, 2020).

(26) CAm=∑i=1MBAS∖M

where M is the size of the population, and BAS indicates the characteristic value of the inter-antibody affinity.

When the inter-antibody affinity Sv,s>ε is present, the Bl,v = 1. Otherwise Bl,v=0. where ε is a predefined threshold value. Excitation degree is the final evaluation result of antibody quality, which is usually obtained by mathematical operation of antibody affinity Sv,s(i) and antibody concentration C(Am). The function of excitation degree is as follows in Eq. (27).

(27) SimAS(Am)=τ×Sv,s(i)−ℓ×C(Am)

where τ,ℓ indicates the affinity and concentration adjustment factors.

The matching between location scheme and optimization problem can stimulate the generation of new location schemes (Bin, Zhao & Yang, 2018). The probability that an individual is selected for a mutation operation is the expected reproduction probability in Eq. (28).

(28) PS(Am)= Γf(Sjl)∑f(Sjl)+(1− Γ )CAM∑CAM

where Г is the constant range from 0 to 1. The greater the affinity between antibody and antigen f(Sjl), the greater the expected reproduction probability PS and the greater the possibility of being selected as a mutant. The higher the antibody concentration C(Am), the lower the expected reproduction probability PS and the lower the possibility of being selected as a mutant individual.

Improving operator

IA is usually insufficient ability to maintain population diversity and easy to fall into local optimization, which leads to reduce convergence speed and optimization performance greatly accompanied by site selection scale. In this article, clone variation, variable threshold selection operator and population update are adopted to preserve the self-regulation mechanism of the original immune optimization algorithm, which is helpful to increase the diversity of location schemes, improve the global optimization ability of the algorithm, and enhance the accuracy of the late solution of the algorithm to achieve rapid convergence.

Each antibody in the memory bank is cloned, the cloning scale is Bc, and each clone copy is mutated according to probability pm. The clone reproduction function B(Am) is calculated as follows in Eq. (29).

(29) B(Am)=round [ζAm\(CAM∑jUf(Sjl))]

where ζ is the cloning factor, f(Sjl) is the value of the affinity function of the antibody, CAm is the value of the antibody Am concentration value, ∑jUf(Sjl) is the sum of antibody affinity values in the clone parent population, and U is the clone parent population size, round(·) is the integer function.

The immune selection operator can determine which antibodies can enter the clonal selection operation based on the expected selection probability (Xiong & Gong, 2019). Setting different thresholds is conducive to improving the solving efficiency according to different search periods (Yoo & Hajela, 1999). For this purpose, antibody concentration is changed in stages by a piecewise function, the equation is as follows in Eq. (30).

(30) T={a,ift≤taveasin⁡(π2×tmax−ttmax−tave),ift>tave

where ϑ is the threshold selection factor. t indicates the current iteration number. tave is the average of evolutionary generations. tmax indicates the maximum iteration number.

The first antibodies Np with better affinity and the re-initialized Popsize−Np antibodies were selected to form a new generation population. According to the three modes of service coverage proposed in Section 1, some location points of the diversion track may be covered by two or more flight service stations at the same time, as shown in Fig. 7.

Figure 7 Schematic diagram of the joint service coverage of the two sites.

Ptk(x,y,z) indicates that the location belongs to joint coverage service. dtik and dtjk indicate the distance between the location and two adjacent service sites, respectively. In this article, the signal space masking is not considered for the time being. Adjusting the service level of candidate sites, changing their service radius, and re-calculating their respective service distances, the affinity value of the IA is determined in Eq. (31).

(31) Sv,sl(i)=rilril+dtik

Implementation flow and algorithm complexity

MIA in this article adopts the group search strategy and the optimal solution with high probability is got through iterative calculation (Wang, 2017). The implementation process of the MIA is shown in Fig. 8.

Figure 8 Process of iterative search algorithm.

The MIA in this section consists of three main modules (Yang et al., 2019), which are antigen recognition and initial antibody generation, intrinsic immune evaluation and adaptive immune manipulation.

The solution steps are description from 1st to 8th step.

1st Step: Antigen identification. According to the set of low-altitude transit flight regions Ωd=[ω1d,ω2d...ωkd...ωKd], establish the time and position of track coordinates for any transit corridor k respectively, convert the transit corridor flight into coordinate relay, construct the affinity function dtk(ij) and various constraints.

2nd Step: Generate the initial population. Based on the actual flight trajectory coordinates, determine all the candidate station solutions that satisfied dtik≤ril around the t-trajectory at any moment Am=(S1l,S2l...Sjl...SNl).

3rd Step: Affinity evaluation. The affinity functions f(Sjl) and Sv,s(i) are called to evaluate the antibody-antibody affinity eigenvalues BAS in the population Sv,s.

4th Step: Inherent immunity termination judgment. Judge whether the weight coefficients of each objective function τ1,τ2,τ3andτ4 satisfy the termination condition, if so, the algorithm terminates, turn to 8th step. otherwise, turn to 5th step to continue the second layer of immune search.

5th Step: Calculation of antibody concentration and excitation degree. According to the principle of lowest rank l configuration of candidate sites and minimum service distance priority mindtk(ij), based on the concentration of antibody C(Am) and the degree of excitation SimAS(Am).

6th Step: Adaptive immune processing. Including the immune selection threshold T(m), clonal propagation B(Am), variation probability Ps(Am), to determine the antibody with high excitation is selected to enter the clonal selection operation (Chen & Zhang, 2023).

7th Step: Population refresh. Using a change in site level, the population update function Sv,sl(i) is called to replace the less motivated antibodies in the population with the generated new antibodies, and determine whether a new generation of antibodies is formed, and if so, go to 6th step to continue the search for superiority. otherwise, go to next step.

8th Step: Site selection decision scheme output. Output the location, number and grade of the final site, and call the objective calculation functions f1(x), f2(x), f3(x) and f4(x) respectively to get the indicators of the optimal solution.

Considering the population size assumed in Problem description, there are k transition flight corridors. Each flight corridor has Ptk location points, and there are N candidate service sites. However, according to the service coverage requirements of each transition flight corridor k, only site Nk that dtik is less than or equal rtil can be included in the search scope.

According to the rules for calculating algorithm complexity (Qian, 2017), the total time complexity of executing one generation of search for the immune search algorithm for this site selection is shown in Eq. (32).

(32) T(n)=2O(Nk/2)+3O(Nk)+O(3Nk/2)+O((2Nck))2+O(2Ncklog⁡(2Nck))

where Nk denotes the number of candidate sites around the kth transit flight path. Nkc (Nkc ≤ Nk) means the number of candidate sites after immune siting and ranking.

Example analysis

Demand characteristics

This article take the service coverage of low-altitude transition corridor flight in Sichuan as the research object to solve and simulate the multi-objective double-layer immune location decision. The results are compared with the traditional dynamic programming and other intelligent search algorithms to verify the feasibility and optimization effect of the model and algorithm. The MIA for LAFSS is simulated by the MATLAB platform. The final visualization of location decision is carried out based on the Spring open source application framework of Java platform. Easyui is used to obtain Cesiumjs map data interface through the front-end development, and the site selection scheme is presented with the terrain data of the map of the world as the background.

Sichuan Province is one of autonomous flight areas to reform the coordinated management of low altitude airspace in China, and has explored and practiced the integrated operation mode of low altitude airspace based on visual autonomous flight. By the end of 2022, 385,378 general aviation flights and 114,606 h have been achieved (Hu, 2021). The Low Altitude Airspace Cooperative Operation Center of Sichuan Province which was completed in 2018 has formulated regulations on the coordinated management and use of low altitude airspace.

Reference to “Sichuan General Aviation Industry Development Plan (2019–2025)”, the implementation strategy of general aviation industry will create a core-two wings, multi-point N network (Sichuan Provincial Development and Reform Commission, 2017). According to the policy planning and development strategy, this article selects the east-west and north-south transfer flight corridors respectively. The schematic diagram of track layout is shown in Fig. 9.

Figure 9 Diagram of space requirements and candidate sites.

Low altitude in Sichuan has set up five cooperative management airspace and five low altitude visual flight corridors, a total of more than 6,600 km2 of pilot airspace (Sichuan Province National Defense Science Technology and Industry Office, 2018). The two transfer flights selected in this article span four regions which are with large differences in terrain change and economic development. The north-south transfer is mainly used for navigation training, industrial and agricultural operations and sightseeing business. The east-west transfer is a high-altitude flight, which undertakes the task of emergency rescue. The codes of main requirement identification points on the two transfer paths are shown in Table 1.

Table 1 Coordinates and codes of demand identification points.

Path 1 ω1d	Path 2 ω2d	
No.	Name	Coordinates	Demand coding	No.	Name	Coordinates	Demand coding	
1	Guangyuan Panlong Airport	105.89	Ps1(x,y,z)	1	Nanchong Yingshan Airport	106.11	Ps2(x,y,z)	
		32.64				30.781		
2	Jiange Airport	105.52	P11(x,y,z)	2	Suining Airport	105.45	P12(x,y,z)	
		32.28				30.35		
3	Jiangyou Airport	104.72	P21(x,y,z)	3	Lezhi Airport	105.02	P22(x,y,z)	
		31.75				30.2		
4	Mianyang Nanjiao Airport	104.56	P31(x,y,z)	4	Ziyang Airport	104.6	P32(x,y,z)	
		31.53				30.11		
5	Dujiangyan Airport	103.64	P41(x,y,z)	5	Renshou Airport	104.13	P42(x,y,z)	
		30.98				29.99		
6	Qionglai Airport	103.46	P51(x,y,z)	6	Pujiang Airport	103.50	P52(x,y,z)	
		30.41				30.19		
7	Ya’an Airport	103.10	P61(x,y,z)	7	Baoxing Airport	102.81	P62(x,y,z)	
		30.06				30.36		
8	Hanyuan Airport	102.65	P71(x,y,z)	8	Xiaojin Airport	102.36	P72(x,y,z)	
		29.34				30.99		
9	Mianning Airport	102.17	P81(x,y,z)	9	Jinchuan Airport	102.06	P82(x,y,z)	
		28.55				31.47		
10	Xichang Qingshan Airport	102.26	P91(x,y,z)	10	Malcolm Airport	102.20	P92(x,y,z)	
		27.89				31.90		
11	Dechang Airport	102.17	P101(x,y,z)	11	Luhuo Airport	100.67	P102(x,y,z)	
		27.40				31.39		
12	Miyi Airport	102.11	Pe1(x,y,z)	12	Ganzi Gesser Airport	101.96	Pe2(x,y,z)	
		26.890				30.04		

The coordinate conversion method described in Coordinates conversion section is used to convert geography to Cartesian. Based on the location Ptk(x,y,z) of every track at any time t, all candidate stations that meet the service distance corresponding are searched nearby the current station level.

Coding

According to the data preparation requirements of IA, all the candidate sites with class A and Class B in the vicinity of two diversion flight paths is coding. The encoding results are shown in Table 2. There are 84 candidate sites, which includes 13 existing airports and 71 planned airports. The candidate sites are classified five parts according to topographic changes. The Class B flight service stations that have been built are Zigong and Guangyuan respectively, and the initial allocation in coding is the same as that of other candidate stations, and the grade is assigned to be four.

Table 2 Codes of candidates sites in Sichuan Province.

Region	No.	Name	Type	Codes of candidate	Region	No.	Name	Type	Codes of candidate	
Chengdu Plain	1	Xindu	In 5 years	S1	Eastern
Sichuan	43	Zigong Rong	In 10 years	S43	
2	Dujiangyan Qingcheng	Heliport	S2		44	Luzhou Hejiang	In 10 years	S44	
3	Longquanyi Hiodai	Heliport	S3		45	Luzhou Xuyong	In 10 years	S45	
4	Pengshanj Iangkou	Heliport	S4		46	Neijiang Weiyuan	In 10 years	S46	
5	Jintang General Airport	Construction	S5		47	Yibin Pingshan	In 10 years	S47	
6	Shuangliu	In 5 years	S6	Southern
Sichuan	48	Leshan Shizhong	In 10 years	S48	
7	Chongzhou Tiangong	Heliport	S7		49	Baoxing	In 5 years	S49	
8	Pidu	In 5 years	S8		50	Ebian	In 5 years	S50	
Northern
Sichuan	9	Guangyuan Panlong	Transport Airport	S9		51	Pujiang	In 5 years	S51	
10	Mianyang Beichuan	Heliport	S10		52	Hanyuan	In 5 years	S52	
11	Beichuan General Airport	Construction	S11		53	Mabian	In 5 years	S53	
12	Mianyang Baiyun	Heliport	S12		54	Yaan	In 10 years	S54	
13	Guanghan High Tech Zone	Heliport	S13		55	Baoxing	In 5 years	S55	
14	Deyang Puxin	Heliport	S14		56	Xichang	In 5 years	S56	
15	Deyang General Airport	Construction	S15		57	Shimian	In 5 years	S57	
16	Shifang General Airport	Construction	S16		58	Ebian	In 5 years	S58	
17	Jiangyou	In 5 years	S17		59	Mabian	In 5 years	S59	
18	Yanting	In 5 years	S18		60	Danling	In 10 years	S60	
19	Pingwu	In 5 years	S19		61	Panzhihua Miyi	In 5 years	S61	
20	Wanyuan	Construction	S20		62	Liangshan Muli	In 10 years	S62	
21	Qingchuan	In 5 years	S21		63	Mianning	In 10 years	S63	
22	Cangxi	In 5 years	S22		64	Leibo	In 10 years	S64	
23	Guangan Linshui	In 5 years	S23		65	Liangshan Ningnan	In 10 years	S65	
24	Dazhou Yihan	In 5 years	S24		66	Liangshan Zhaojue	In 10 years	S66	
25	Dazhou Qu	In 5 years	S25	Western
Sichuan	67	Malcolm Fuhua	Heliport	S67	
26	Bazhong Nanjiang	In 5 years	S26		68	Abajinchuan	In 5 years	S68	
27	Guangyuan Wangcang	In 10 years	S27		69	Aba Xiaojin	In 5 years	S69	
28	Nanchong Xichong	In 10 years	S28		70	Aba Heishui	In 5 years	S70	
29	Guangan Huaying	In 10 years	S29		71	Malcolm	In 5 years	S71	
30	Yingshan	In 5 years	S30		72	Aba	In 5 years	S72	
31	Bazhong Pingchang	In 10 years	S31		73	Ruoergai	In 5 years	S73	
Eastern Sichuan	32	Lezhi	In 10 years	S32		74	Ganzi Jiulong	In 5 years	S74	
33	Fengming General Airport	Runway	S33		75	Daofu	In 5 years	S75	
34	Anyue	In 5 years	S34		76	Dege	In 5 years	S76	
35	Shehong	In 5 years	S35		77	Shiqu	In 5 years	S77	
36	Yanyuan Shuanghe	Heliport	S36		78	Seda	In 5 years	S78	
37	Yanyuan General Airport	Construction	S37		79	Litang	In 5 years	S79	
38	Zigong Fushun	In 5 years	S38		80	Aba li	In 5 years	S80	
39	Luzhou Gulin	In 5 years	S39		81	Mao	In 5 years	S81	
40	Yibin Jiangan	In 5 years	S40		82	Kangding	In 10 years	S82	
41	Xingwen	In 5 years	S41		83	Xinlong	In 10 years	S83	
42	Junlian	In 5 years	S42		84	Baiyu	In 10 years	S84	

By encoding, a scheme Am is formed, i.e., a solution. The total encoding length N of the solution is 84, i.e., the number of candidate sites Fig. 10. The part of the solution ωkd represents the number of the requirement point that the solution satisfies. The m part of the solution represents the candidate site label j. The s part of the solution indicates whether or not the corresponding candidate site is selected and the level of selection.

Figure 10 Coding representation of the solution.

ωkd indicates that the solution meets the requirement. The candidate site NO.1 has a service station level of 4. Candidate site NO.2 was not selected. The candidate site NO.4 has a service station level of 3. The candidate site NO.73 has a service station level of 1. The candidate site NO.74 has a service station level of 2.

Parameter taking

The sites with level 3 and level 4 layout are mainly selected in the scope of the example in this article. The required infrastructure construction, production and operation, post-maintenance, and reservation costs are random numbers between 5 to 3, 3 to 2, 2 to 1, and 1 to 0.5 million, respectively. It is also assumed that costs other than infrastructure are proportional to operating time. The costs for each candidate site are shown in Table 3.

Table 3 Cost of candidate sites (Unit: 10 thousand RMB).

Station level
L	Cj1	Cj2	Cj3	Cj4	First year
CL	Adjust
coefficient	Total cost range
CL	
1	900	500	300	200	1,900	±10%	1,800~2,000	
2	600	300	180	120	1,200	±15%	1,000~1,200	
3	400	200	120	80	800	±10%	720~880	
4	300	150	100	50	600	±20%	480~720	

The four key parameters in the algorithm are memory capacity overbest, expected reproduction probability parameter ps, mutation probability pm, and concentration threshold. Suppose that the population size of the algorithm is sizepop = 50, the number of iterations is 600, and other parameters are shown in Table 4.

Table 4 Parameter levels.

Parameters	Level	
1	2	3	4	
Overbest	6	8	10	12	
Pm	0.5	0.6	0.7	0.8	
Ps	0.8	0.85	0.9	0.95	
ε	0.4	0.5	0.6	0.7	

Table 4 shows four reasonable level values for the four key parameters of the experiment, with each pair of parameters being run 20 times.

Dynamic programming site selection based on service distance

The dynamic programming (DP) algorithm (Annear, Akhavan-Tabatabaei & Schmid, 2023) sets the general aviation airports that the transition flight track passes through as node Ptk(x,y,z), determines the distance between each node and the nearby candidate stations according to the screening condition dtik≤rsil of effective service distance, and obtains the dynamic programming path with service radius as the cost. Taking the north-south transition low-altitude general navigation flight as an example, the minimum cost dynamic programming graph as shown in Fig. 11 is obtained according to the distance of the candidate stations around the track.

Figure 11 Distance cost of service response path.

The path from node P1i to P1j is unidirectional and bidirectional, and the path without arrow is bidirectional, which provides a variety of options. Starting from P1S, according to the 10 important nodes experienced by the flight trajectory, the service distance was calculated in stages, and the optimal location and service allocation scheme under this method was finally selected. The results of the dynamic site selection are shown in Fig. 12.

Figure 12 Location sites of dynamic plan.

The number of sites selected by the dynamic programming scheme is eight, including one in the region and seven in class B. According to the principle of minimum cost and maximum flow, a regional LAFSS is used to cover Chengdu Plain, north Sichuan and east Sichuan.

However, its utilization seems to be extremely high, but the quality of service is affected in many areas due to signal blocking. Therefore, in this article, the inherent and adaptive characteristics of double-layer immunity, dynamic selection and clonal mutation, flexible adjustment of the level, find the optimal scheme.

MIA site selection optimization with two-layer response

Two demand corridors are set for transition flight, and each corridor involves 12 key turning points. The number of candidate sites is 84, which are two Class B flight service stations, 16 general aviation airports, and 68 cities and towns with planned general aviation airports. The level initialization of all candidate stations is set to four, and the service radius is set to 100 km. According to the safe flight criterion that the flight demand is covered by the service, the initial population is obtained as shown in Fig. 13.

Figure 13 Generation of initial populations.

The initial population consisted of 14 flight service stations, including three in Chengdu Plain, two in East Sichuan Hills, two in north Sichuan, three in south Sichuan and four in West Sichuan Plateau. Xindu in Chengdu Plain, weiyuan in southeast Sichuan and yingshan in northeast Sichuan have higher service efficiency. Except for qingchuan in the northwest of Sichuan, there are two service stations, and the remaining 10 stations all cover a navigable airport. Therefore, the repeated coverage areas are more in north Sichuan, Chengdu Plain and east Sichuan hills.

Innate immunity is a natural immune defense function formed during the development of the body. Zigong Fengming Airport S33, as an existing Class B flight service station, is compared to the non-specific defense function that has been available since birth. It is used as an immune body to search for germline development and evolution. In this article, according to the cost fluctuation of site selection, the auto-immune search was carried out according to the service radius of 120 and 150 km respectively. Finally, the innate immunity site selection schemes P1IA and P2IA were obtained respectively, as shown in Table 5.

Table 5 Inherent immunization site selection options.

P1IA	P2IA	
Station	Longitude	Latitude	Including airport	Station	Longitude	Latitude	Including airport	
Jianyang	104.53	30.38	19	Lezhi	105.02	30.3	23	
Yilong	106.38	31.52	6	
Pingwu	104.52	32.42	3	Langzhong	105.97	31.75	5	
Neijiang	105.04	29.59	2	
Ningnan	102.76	27.07	2	Jiulong	101.53	29.01	3	
Lushan	102.91	30.17	1	Hongyaun	102.55	31.79	2	
Maerkang	102.22	31.92	1	Miyi	102.15	26.9	1	
Luhuo	100.65	31.38	1	Neijiang	105.04	29.59	1	
Litang	100.28	30.03	1	Rangtang	100.97	32.3	1	

The comparison of the two site selection schemes is known, the quantity of service station is different by two Class B, the cost is increased by 22.22%, the maximum service demand ratio P2IA is increased by 11.11%, and the equilibrium P1IA is increased by 11.43%. Based on Minitab Statistical Software platform, this article used the Taguchi experimental design method to discuss parameter setting and sensitivity analysis. The horizontal trend of the influence factors shown in Fig. 14 was obtained by using multivariate analysis of variance.

Figure 14 Trend of impact factors.

Therefore, other relevant parameters of the algorithm were set according to the horizontal trend in Fig. 15. Population size sizepop = 50, memory bank capacity overbest = 8, expected reproduction probability parameter ps = 0.9, mutation probability pm = 0.7, concentration threshold ε = 0.7.

Figure 15 The value of the sign change after the special normalization of each index.

Due to the initial effect of innate immunity with invasive antigen substances, the demand for flights in the western Sichuan region is low, so that the number of stations in this region for both schemes is only one. In order to avoid the disadvantages of poor compressive strength and no specific selectivity, adaptive immunity needs to be introduced. Thus, the advantages of randomness, parallelism, global convergence and population diversity are utilized to obtain an optimized location decision scheme.

The location of LAFSS in Sichuan Province can be considered as containing five main antigens from east, west, north, south and Chengdu Plain respectively, and multiple sub-antigens and antibody groups can be decomposed from each region through adaptive adjustment. In this way, more excellent antibodies can be provided for future high-level immunity, and multiple extremes or global equilibrium optimal search characteristics can be played. The change characteristics of Sv, s(i) and f(Slj) are obtained in Fig. 15.

In order to show the comparison of each index function more intuitively, this article adopts normalization processing. The term of the maximum value of the current objective function is set as one, and the rest is taken in this proportion.

The statistical characteristics of economic cost ratio are 1.49 in Chengdu Plain, 1.59 in north Sichuan, 1.3 in east Sichuan, 1.83 in south Sichuan and 2.29 in west Sichuan. The highest service response ratio is 2.83 in western Sichuan, which is 3.5 times higher than the lowest of 0.89 in Chengdu Plain. The highest proportion of repeated coverage is 2.6 in western Sichuan, which is 25 times higher than the lowest in Chengdu Plain. The highest effective coverage ratio is 1.95 in Chengdu Plain, which is 2.5 times higher than 0.76 in western Sichuan Plateau. However, the highest Sv, s(i) and f(Slj) belong to the south and east regions of Sichuan where all costs are more balanced, respectively.

According to the variation rules of Sv,s(i) and f(Slj), cloning will continue to be used to reproduce the above two regional good genes. After repeated coverage of mutations and selection, Chengdu Plain, as the center of southwest China, has been established as an excellent gene, four site selection indicators are well balanced, and total affinity is the best. The other two candidate sites for expansion and upgrading are “yuexi” in southern Sichuan and “aba” in western Sichuan. After 600 iterations, the mating probability is 0.6 and the mutation probability is 0.1. The decision scheme and service allocation as shown in Fig. 16 are obtained through the simulation platform with MIA.

Figure 16 Adaptive immunization decision scheme and service allocation.

According to the dynamic programming decision in Fig. 12 and the site selection results completed in this section, the index comparison of the three algorithms is obtained. The comparison results are shown in Table 6.

Table 6 Comparison of location results with different algorithm.

	Existing scheme	Dynamic plan	Common IA	Multi-level
IA	Modified results
(in %)	
Z1 (in 10 thousands)	1,200	4,369	3,250	2,390	−25.45/−26.46	
Z2 (in seconds)	>180	75	68	51	−9.33/−25	
Z3 (in km2)	62,800	160,768	101,627	71,592	−36.78/−29.5	
Z4 (in km2)	Null	341,632	251,623	305,208	+35.7/+17.5	
Iterations	Null	500	450	320	−36	
Most affinity value	Null	2.01	1.94	1.82	+9.5	

The optimization effect of the MIA on the location decision of LASS in Sichuan Province is as follows: the cost index Z1 is reduced by 26.46%, the service response is increased by 25%, the repeated coverage is reduced by 29.5%, and the effective service coverage area is increased by 17%. In terms of the time complexity of solving the problem, the iteration efficiency was increased by 36%, and the total immune affinity was increased by 9.5%.

Conclusions

Location decision of LAFSS is combination optimization problems with multi-objective constraints. MIA is put forward from the perspective of service coverage and flight cost in the way of service relay. The visualization of location decision is realized by using artificial intelligence algorithm and computer graphics technology. The main conclusions are given as following.

Based on kinematic model of the object, the equation of simple aviation straight track and precise position of low-altitude flight and transfer flight is constructed, and the scope of service demand is accurately defined.

The constrained multi-objective optimization mathematical model established satisfies the basic criteria of LAFS safety, takes into account the principles of economy of low-altitude flight industry and service efficiency, and the conflict analysis between objectives is helpful for scientific location decision.

The proposed two-layer response and the Taguchi experimental design obtained an improved IA with the best combination of operating parameters. By using the innate immunity to retain the good genes, the function of adaptive adjustment is played, and the balance and optimal of each goal is achieved. The time complexity of the algorithm is effectively reduced through the above improvements. The MIA for LAFSS is given out with Matlab simulation as shown in Fig. 17, which the three red circles are the most optimal decision.

Figure 17 Iterations and final locations.

The final visualization of location decision is carried out based on the Spring open source application framework of Java platform.

Location selection of a LAFSS in Sichuan Province is taken as an example to verify the model and MIA. The various indicators comparison of decision schemes with the three algorithms is given out in Fig. 18. Easyui is used to obtain Cesiumjs map data interface through the JAVA development, and the site selection scheme is presented with the terrain data of Map of the world as the background in Fig. 19.

Figure 18 Comparison of siting options for inherent and adaptive immunity.

Figure 19 Comparison of cost indicators corresponding to the three algorithms.

The location problem of LAFSS is a comprehensive decision. It not only needs to meet the relevant provisions of low-altitude airspace management, but also consider various limiting factors such as terrain characteristics on the signals of communication and navigation facilities, economic and industrial structure between regions and travel behavior differences. It is also necessary to plan for the special situation of emergency rescue. Therefore, the layout of LAFSS is a multi-objective planning work with increasing constraints. With the expansion of site selection scale, scientific determination of site selection objectives and intelligent search algorithms ensure flight safety and take into account the sustainable development of the industry.

Supplemental Information

Supplemental Information 1 Code.

Click here for additional data file.

Supplemental Information 2 The best combination of operating parameters are obtained by proposing two-layer response and Taguchi experimention on Matlab simulation platform.

As the times of iterations increases, the value of objective function decreases gradually. When iterated upto 1000 times, the objective function no longer changes. The cost of site selection is proportional to the number of service stations.The blue circles represent candidate sites. The red circles are the sites that are selected finally.

Click here for additional data file.

Supplemental Information 3 Taking the site selection of a low-altitude flight service station in Sichuan Province as an example, the Spring open source application framework based on the Java platform is developed through the front-end development of easyui and cesiumjs map data in.

The range of white circle indicates service area of flight service station. The center point of circle is the coordinates of general airport or landing point where the flight service station is located. The intersection area means service coverage repeated.

Click here for additional data file.

Additional Information and Declarations

Competing Interests

Author Contributions

Data Availability

The authors declare that they have no competing interests. Tang Xie is employed by Sichuan Highway planning Survey Design Institute Co. LTD.

Huaqun Chen conceived and designed the experiments, performed the experiments, analyzed the data, performed the computation work, prepared figures and/or tables, authored or reviewed drafts of the article, and approved the final draft.

Weichao Yang conceived and designed the experiments, performed the experiments, analyzed the data, performed the computation work, prepared figures and/or tables, and approved the final draft.

Xie Tang analyzed the data, prepared figures and/or tables, map data, and approved the final draft.

Minghui Yang conceived and designed the experiments, performed the experiments, analyzed the data, performed the computation work, prepared figures and/or tables, and approved the final draft.

Fangwei Huang performed the computation work, prepared figures and/or tables, and approved the final draft.

Xingao Zhu performed the experiments, prepared figures and/or tables, and approved the final draft.

The following information was supplied regarding data availability:

The modified immune algorithm simulated in MATLAB and the visualization presentation of location schemes with the Spring framework of the JAVA platform are available in the Supplemental Files.

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
