# Peer review of "Location decision of low-altitude service station for transfer flight based on modified immune algorithm"

_PeerJ Computer Science, doi:10.7717/peerj-cs.1624_

## Round 0.1 · original submission · Major Revisions

Both reviewers have offered insightful and high-quality comments, and the authors should meticulously address these valuable suggestions during the manuscript revision process. It is essential to focus on enhancing the overall structure of the paper. To achieve this, the Introduction section should be revised to emphasize the significant contributions of this work. Additionally, the related works currently located in the Introduction should be appropriately relocated to their respective sections. Furthermore, the authors are encouraged to include more recent literature to strengthen the research background and provide up-to-date support for their study.

**Language Note:** PeerJ staff have identified that the English language needs to be improved. When you prepare your next revision, please either (i) have a colleague who is proficient in English and familiar with the subject matter review your manuscript, or (ii) contact a professional editing service to review your manuscript. PeerJ can provide language editing services - you can contact us at copyediting@peerj.com for pricing (be sure to provide your manuscript number and title). – PeerJ Staff

Reviewer 1 ·

Basic reporting

The manuscript addresses the complex problem of low-altitude flight service station location selection and solves it using an improved immune algorithm. The method presented in the article is reasonable and well-structured, but there are the following issues:
(1) The problem that the manuscript aims to solve should be clearly and explicitly stated in the introduction part.
(2) What are the contributions of the manuscript compared to existing research? These should also be presented in the introduction part.
(3) The complexity analysis of the algorithm in section 4.3.2 indicates that the algorithm has an exponential complexity. Is this complexity considered too high? It might be worth considering whether to remove this subsection.
(4) In Section 5.3, are the solutions obtained through dynamic programming feasible or infeasible? If they are infeasible, including this part might not be meaningful. If they are feasible, the efficiency and quality of the solutions need to be quantitatively explained.
(5) In the case study, the detailed step-by-step computation process of the algorithm does not need to be presented.
(6) In the conclusion, it is mentioned that three algorithmic location decision schemes and various indicators are compared. It is recommended to add a comparative table of these indicators to provide a more intuitive presentation.

Experimental design

no comment

Validity of the findings

no comment

Reviewer 2 ·

Basic reporting

INTRODUCTION

(1) Algorithm description paragraph should be moved to the algorithm design section.

(2) There are few related studies abroad, which need to be increased appropriately.

Parameter setting Section:

(1) The title is suggested to be changed to be model assumptions, and related content should be added in this section.

(2)Describe all parameter symbols appearing in the article.

TRANSITION TRACK

(1)Streamlining is recommended.
(2)Direct focus on trajectory determination method adopted in this paper.

Mathematical model Section

(1) The title of "Mathematical Model" is suggested to be changed to "Objective function".

(2) A separate section is recommended to describe each category of constraints.

Experimental design

IA for MULTI-LAYER RESPONSE

(1) The principle of immune algorithm should be greatly simplified.

(2) The solution process of the site selection optimization part of the improved immune algorithm with two-layer response is too detailed, so it is suggested to simplify it appropriately.

Validity of the findings

EXAMPLE ANALYSIS
(1)A new section should be added to describe the advantages of the improved algorithm compared with other traditional algorithms or solving tools such as CPLEX.

(2)The Figure of low altitude coordinated airspace in Sichuan is poor in definition.

Additional comments

Enhance the labeling details of variables
Check the syntax of the full text
The format of the chart

---

## Round 0.2 · accepted · Accept

The authors address the reviewers' concerns and recommend to accept this paper. I also suggest the authors adjust Figures 1, 2, 5, and 7 to avoid overlapping fonts, enhance the clarity of Figures 9, 12, 13, 16, 17, and 18, and increase the font size of Figure 19. Keywords should also be adjusted to describe the background and technology rather than the names of frameworks such as Java and Matlab.

Reviewer 1 ·

Basic reporting

Thank you for the author's patient responses. All of my questions have been addressed and the necessary revisions have been made. The current version of the paper meets the publication standards.

Experimental design

no comment

Validity of the findings

no comment

Reviewer 2 ·

Basic reporting

no comment

Experimental design

no comment

Validity of the findings

no comment

Additional comments

no comment